# Health disparities in cervical cancer: Estimating geographic variations of disease burden and association with key socioeconomic and demographic factors in the US

Tara Castellano[1‡], Andrew K. ElHabr[2‡*], Christina Washington[3], Jie Ting[4], Yitong J. Zhang[4], Fernanda Musa[5], Ezgi Berksoy[2], Kathleen Moore[4], Leslie Randall[6], Jagpreet Chhatwal[7], Turgay Ayer[8,9‡], Charles A. Leath, III[10‡]

1 Department of Gynecologic Oncology, Louisiana State University, New Orleans, Louisiana, United States of America, 2 Value Analytics Labs, Boston, Massachusetts, United States of America, 3 Department of Obstetrics and Gynecology, Stephenson Cancer Center, Oklahoma City, Oklahoma, United States of America, 4 Pfizer Inc., Bothell, Washington, United States of America, 5 Swedish Cancer Institute, Seattle, Washington, United States of America, 6 Department of Obstetrics and Gynecology, Virginia Commonwealth University, Richmond, Virginia, United States of America, 7 Massachusetts General Hospital Institute for Technology Assessment, Harvard Medical University, Boston, Massachusetts, United States of America, 8 Department of Industrial and Systems Engineering, Georgia Institute of Technology, Atlanta, Georgia, United States of America, 9 Department of Medicine, Emory University, Atlanta, Georgia, United States of America, 10 Department of Obstetrics and Gynecology, University of Alabama at Birmingham, Birmingham, Alabama, United States of America

‡ TC and AKE are joint first authors. CAL and TA are joint senior authors.
* andrewelhabr@valueanalyticslabs.com

## Abstract

### Background

Despite advances in cervical cancer (CC) prevention, detection, and treatment in the US, health disparities persist, disproportionately affecting underserved populations or regions. This study analyzes the geographical distribution of both CC and recurrent/metastatic CC (r/mCC) in the US and explores potential risk factors of higher disease burden to inform potential strategies to address disparities in CC and r/mCC.

### Methods

We estimated CC screening rates, as well as CC burden (number of patients with CC diagnosis per 100,000 eligible enrollees) and r/mCC burden (proportion of CC patients receiving systemic therapy not in conjunction with surgery or radiation), at the geographic level between 2017–2022 using administrative claims. Data on income and race/ethnicity were obtained from US Census Bureau's American Community Survey. Brachytherapy centers were proxies for guideline-conforming care for locally advanced CC. Associations among demographic, socioeconomic, and healthcare resource variables, with CC and r/mCC disease burden were assessed.

**Data Availability Statement:** The data used to generate the findings in this study can be found at https://github.com/valueanalyticslabs/geoanalyzer-public.

**Funding:** This study was funded by Seagen Inc., which was acquired by Pfizer Inc., in December 2023, and Genmab A/S. The funders had no role in study design, data collection and analysis, decision to publish, or preparation of the manuscript.

**Competing interests:** Tara Castellano has received consulting fees from Glaxo-Smith-Klein; Andrew K. ElHabr and Ezgi Berksoy are paid employees of Value Analytics Labs, a healthcare consultancy company; Jie Ting and Yitong J. Zhang are employees and stack holders of Pfizer Inc.; Kathleen Moore has participated in data monitoring or advisory boards for Astra Zeneca, Aravive, Alkemeres, Aadi, Blueprint pharma, Clovis, Caris, Duality, Elevar, Eisai, EMD Serono, GSK/Tesaro, Genentech/Roche, Hengrui, Immunogen, Janssen, Lilly, Mersana, Merck, Myriad, Mereo, Novartis, OncXerna, Onconova, SQZ, Tarveda, VBL Therapeutics, Verastem and Zentalis, received support for attending meetings from Astra Zeneca, GSK/Tesaro, and BioNTech, and holds a leadership role with GOG partners; Leslie Randall reports personal fees from Seagen Inc. for an educational webinar at drug launch and speaker's bureau and personal fees from Merck for an unbranded educational video for cervical cancer. Her institute receives research funding for clinical research from Seagen and Merck. She reports personal fees from BluPrint Oncology, PER, CurioScience, Projects in Knowledge, AstraZeneca, Tesaro, Merck, Mersana, Agenus, Rubius Therapeutics, Myriad Genetics, EMD Serono, Genentech/Roche, Seattle Genetics, Novartis, and Eisai, all outside the submitted work; Jagpreet Chhatwal and Turgay Ayer are co-owners of Value Analytics Labs; Charles A. Leath III has received consulting fees from Seagen Inc. for service on Scientific Advisory boards, cervical cancer research funding from Agenus, Rubius Therapeutics, and Seagen Inc., and funding from the NCI UG1 CA23330 and P50 CA098252; Fernanda Musa and Christina Washington have no competing interests to disclose.

## Results

Between 2017–2022, approximately 48,000 CC-diagnosed patients were identified, and approximately 10,000 initiated systemic therapy treatment. Both CC and r/mCC burden varied considerably across the US. Higher screening was significantly associated with lower CC burden only in the South. Lower income level was significantly associated with lower screening rates, higher CC and r/mCC burden. Higher proportion of Hispanic population was also associated with higher CC burden. The presence of ≥1 brachytherapy center in a region was significantly associated with a reduction in r/mCC burden (2.7%).

## Conclusion

CC and r/mCC disparities are an interplay of certain social determinants of health, behavior, and race/ethnicity. Our findings may inform targeted interventions for a geographic area, and further highlight the importance of guideline-conforming care to reduce disease burden.

## Introduction

Cervical cancer (CC) is a disease that is mostly preventable and potentially curable, when diagnosed early [1]. Despite the availability of human papillomavirus vaccination as prevention, as well as routine CC screening and effective treatments for cervical dysplasia, approximately 14,000 new cases of CC and 4,300 CC deaths were estimated to occur among US women as of 2023 [2]. Additionally, striking disparities persist throughout the CC care continuum, leading to differences in CC and recurrent or metastatic CC (r/mCC) burden observed across the US [3, 4].

While healthcare disparity is well studied in prevention of CC [5–7], there is limited literature characterizing the impact of disparity on the variation in disease burden across US communities. Existing data are often focused on individual risk factors such as specific race/ethnic groups or from limited-scope registries [8–10]. To adequately address the needs of patients with CC and r/mCC and mitigate associated health disparities across the care continuum, a holistic understanding of the disparities contributing to disease burden across different regions may be helpful. This knowledge will provide professional groups, healthcare providers, policy decision-makers and patient advocacy groups with the necessary tools to target areas in the US where there is a high need for CC or r/mCC intervention. Previously, we quantified and visualized the geographical distribution of CC and r/mCC burden among commercially insured patients in the US between 2015–2020, lending hypotheses to areas with highest need of disease awareness education, or where access to standard treatments in the curative setting are not adequate, which leads to need for salvage treatment options such as systemic therapy [3].

Presently we aim to explore and generate hypotheses of potential demographic, socioeconomic and behavioral risk factors of observed geographic variation of CC and r/mCC burden in the US. Using contemporary data from a large, representative, administrative claims database and relevant secondary data sources, our objective is to quantify associations between burden of CC and r/mCC diseases with CC screening rates, select demographic and socioeconomic factors, and brachytherapy access at the local geographic level. Beyond identifying areas with the highest disease burden, findings from this study may inform future research direction and targeted healthcare resources and interventions to improve outcomes for people at risk or living with CC.

## Materials and methods

### Data

A total of eight factors covering socioeconomic status, demographic profiles, and health care infrastructure potentially contributing to geographical variation in CC and r/mCC disease burden were considered for our analyses. Of these, four (CC screening, poverty, race/ethnicity, and brachytherapy center availability) were identified to have potentially highest relevance for the study and retained for further analyses, based on prior literature and clinical expert opinion [11–13].

Adult patients with CC and r/mCC, and cervical cancer screening data (either cytology or high-risk human papillomavirus [hrHPV] testing) were identified from administrative claims data in Komodo Healthcare Map (2017–2022). The closed claims dataset includes >165 million US patients covered by Commercial, Medicaid, or Medicare Advantage plans [14].

The US Census Bureau's American Community Survey was used to collect information on income level, in which we define low-income level as a family with income < 200% of the federal poverty limit (FPL), and race/ethnicity at the ZIP-5 level, which were then aggregated to the ZIP-3 level [15]. Mutually exclusive race/ethnicity subgroups were defined as Asian, Black, Hispanic, White and Other (including unidentified). The existence of ≥1 brachytherapy (internal radiation) center in a ZIP-3 was obtained from the American Brachytherapy Society and was used as a proxy for guideline-conforming care for locally advanced CC [16, 17].

### Patient population and definitions

A CC patient was defined as a female ≥18 years old having ≥1 inpatient claim or ≥2 outpatient claims with a diagnosis for malignant neoplasm of the cervix as identified by the ICD-9-CM code 180.xx or ICD-10-CM code C53.xx. An r/mCC patient was defined as a CC patient who initiated systemic therapy listed by the National Comprehensive Cancer Network guidelines for treatment of r/mCC, which were not associated with surgery or radiation [18, 19]. The date of first initiation of systemic therapy in the r/mCC setting is the r/mCC index date. Patients must have ≥6 months of continuous medical and pharmacy enrollment prior to the r/mCC index date, ≥3 months of continuous enrollment after the index date and have no more than 8 claims for a non-specific chemotherapy encounter (ICD-10-CM code J99.99).

A patient was considered screened for cervical cancer during a year of interest if she was either between the ages of 21 and 64 and had cervical cytology performed within the previous three years, or between the ages of 30 and 64 and had cervical hrHPV testing performed with or without cytology within the previous five years [20]. Cervical cancer screening data were available in Komodo Healthcare Map since 2019.

To identify geographic ZIP-3 level distribution of annual CC burden, we calculated the prevalent number of CC diagnosis per 100,000 female enrollees ≥18 years for each year and ZIP-3. The annual r/mCC burden was defined as the number of incident r/mCC cases (initiated systemic therapy) per the number of prevalent CC cases in each year and ZIP-3.

The administrative claims data were processed between February and September 2023 to generate distributions at the ZIP-3 level for research purposes. Patients or enrollees without ZIP-3 information were excluded from analysis. The authors had no access to information that could identify individual patients. ZIP-3 data were censored where patient counts were ≤3. No ethics approval was needed because this is a retrospective study that did not utilize patient-level identification for analyses.

## Statistical analysis

Univariate regressions were used to explore the associations between CC screening rate, race/ethnicity, and household income with regional CC and r/mCC burden, at the ZIP-3 level. We analyzed the association between the availability of a brachytherapy center and CC and r/mCC burden similarly, in which we encoded the presence of at least one brachytherapy center in a ZIP-3 as the binary independent variable of interest. Two-tailed t-tests were used to determine if estimated slopes from the regressions were significantly different from 0. A p-value of <0.05 was considered statistically significant. All statistical analyses were performed using R 4.2.2., in which the packages mapbox, tidycensus and tigris, were used [21–24].

## Geo-analyzer update

We previously developed the Cervical Cancer Geo-Analyzer (http://www.geo-analyzer.org), a publicly available online, interactive tool that allows users to visualize CC and r/mCC disease burden among Commercial-insured beneficiaries, across metropolitan statistical areas (2015–2020) [3]. The currently updated Geo-Analyzer tool includes data on CC and r/mCC burden (2017–2022) from the present study and includes a more representative population comprising Medicare Advantage and Medicaid beneficiaries in addition to the Commercial population. We can display results by the age groups of 18–44, 45–64 and 65+, insurance type, and proportion of households that are ≤200% FPL. Finally, we overlayed brachytherapy center sites, as a proxy for guideline-conforming care for earlier-stage disease, with CC and r/mCC burden.

## Results

Between 2017–2022, the total number of eligible enrollees in the database ranged from 29,343,966 in 2019 to 35,214,693 in 2022 (Table 1). Over this observation period, a total of 48,962 patients had CC diagnoses and of those, 10,898 initiated a systemic therapy treatment (r/mCC patients). Annual burden numbers are delineated in S2 Table. The median age for CC patients in the dataset was 53 years of age while that of r/mCC patients was 59. Approximately three-quarters of the patients in the dataset had privately managed (Commercial or Medicare Advantage) insurance, and one-quarter had Medicaid or managed Medicaid, consistent with a previous report [25]. The geographic variation of CC and r/mCC burden across the US between 2017–2022 at the ZIP-3 level is depicted in Fig 1 and can be visualized in the public online tool. (http://www.geo-analyzer.org). Brachytherapy centers, as a proxy for guideline-confirming treatment for locally advanced disease, are mapped in Fig 1. CC screening rates are visualized in S1 Fig, and the states in each region are outlined in S1 Table.

In the univariate analyses, screening rate was not significantly associated with overall CC or r/mCC burden at the national level (Fig 2). At the regional level however (Fig 2), a higher screening rate was significantly associated with lower CC and r/mCC burden for the South (S3 Table, p<0.001 for CC burden and p = 0.013 for r/mCC burden), and lower r/mCC burden in the Midwest (S3 Table, p<0.01). On the other hand, the West region showed a statistically significant trend between higher screening rate and higher r/mCC burden (S3 Table, p = 0.038).

A higher percentage of low-income households within a ZIP-3 area was significantly associated with higher CC burden overall (Fig 3A, p<0.001). For r/mCC, while the association was not significant at national level, the South region showed a significant increase in burden with an increasing percentage of low-income households (Fig 3B, p = 0.001).

At the ZIP-3 level, two race/ethnicity groups showed significant association with disease CC burden: ZIP-3 with higher proportion of Hispanic population had higher CC burden, and those with higher Asian population had lower CC burden (Fig 4, both p<0.001). However,

**Table 1. Characteristics of eligible enrollees with cervical cancer diagnosis or initiated systemic therapy for recurrent or metastatic cervical cancer in the US (2017–2022).**

| Category | Overall Enrollees | CC Patients | r/mCC Patients |
|---|---|---|---|
| Total | | 48,962 | 10,898 |
| Age, median [IQR][a] | | 53 [42, 63] | 59 [49, 66] |
| Insurance Type | | | |
| *Commercial* | 78.1% | 74.2% | 74.0% |
| *Medicaid* | 21.5% | 25.1% | 24.9% |
| *Other* | 0.4% | 0.7% | 1.1% |
| Region | | | |
| *Midwest* | 22.0% | 21.4% | 20.4% |
| *Northeast* | 19.9% | 21.2% | 22.3% |
| *South* | 38.6% | 38.0% | 37.5% |
| *West* | 19.1% | 18.8% | 19.0% |
| *Other/Unknown* | 0.4% | 0.6% | 0.8% |
| Incident cases by year | | | |
| *2017* | 32,447,904 | 9,907 | 1,764 |
| *2018* | 30,383,147 | 8,953 | 1,798 |
| *2019* | 29,343,966 | 8,371 | 1,843 |
| *2020* | 34,215,828 | 7,481 | 1,797 |
| *2021* | 34,851,039 | 7,846 | 1,853 |
| *2022* | 35,214,693 | 6,404 | 1,843 |

CC, cervical cancer; r/mCC, recurrent or metastatic cervical cancer.

[a] Age statistics are computed over 2015–2022.

when examining by region, we found that this finding by race/ethnicity was mainly driven by Hispanics in the West and Asians in the South.

Also at the regional level, a higher proportion of Black population in a ZIP-3 area was significantly associated with higher CC burden in the Midwest and Northeast but with lower CC burden in the South (Fig 4, S3 Table).

For r/mCC, only the Asian population prevalence in the Midwest was significantly associated with lower burden (Fig 4, p = 0.047). No race/ethnicity group showed significant association with r/mCC burden at national level.

Lastly, we found that the presence of at least one brachytherapy center in a ZIP-3 area was associated with a significant reduction in overall r/mCC burden (Fig 5, 2.7%, p<0.001). In particular, this finding was driven by reductions in the South and Midwest (S3 Table, p<0.001).

## Discussion

Previous disclosures of the pilot version of Geo-Analyzer mapped geographical dispersion of CC and r/mCC burden using limited datasets and were from an era prior to approvals novel therapies in the r/mCC setting [3, 26]. We report in the present study updated data from a comprehensive dataset covering >165 million enrollees across US, with patients in a more contemporary CC and r/mCC treatment landscape. Additionally, although healthcare disparity among US CC patients overall is relatively well-characterized [4, 8, 27–29], we seek to form hypotheses for the first time about how demographic, socioeconomic, or behavioral factors may influence regional differences in CC or r/mCC burden that we observe.

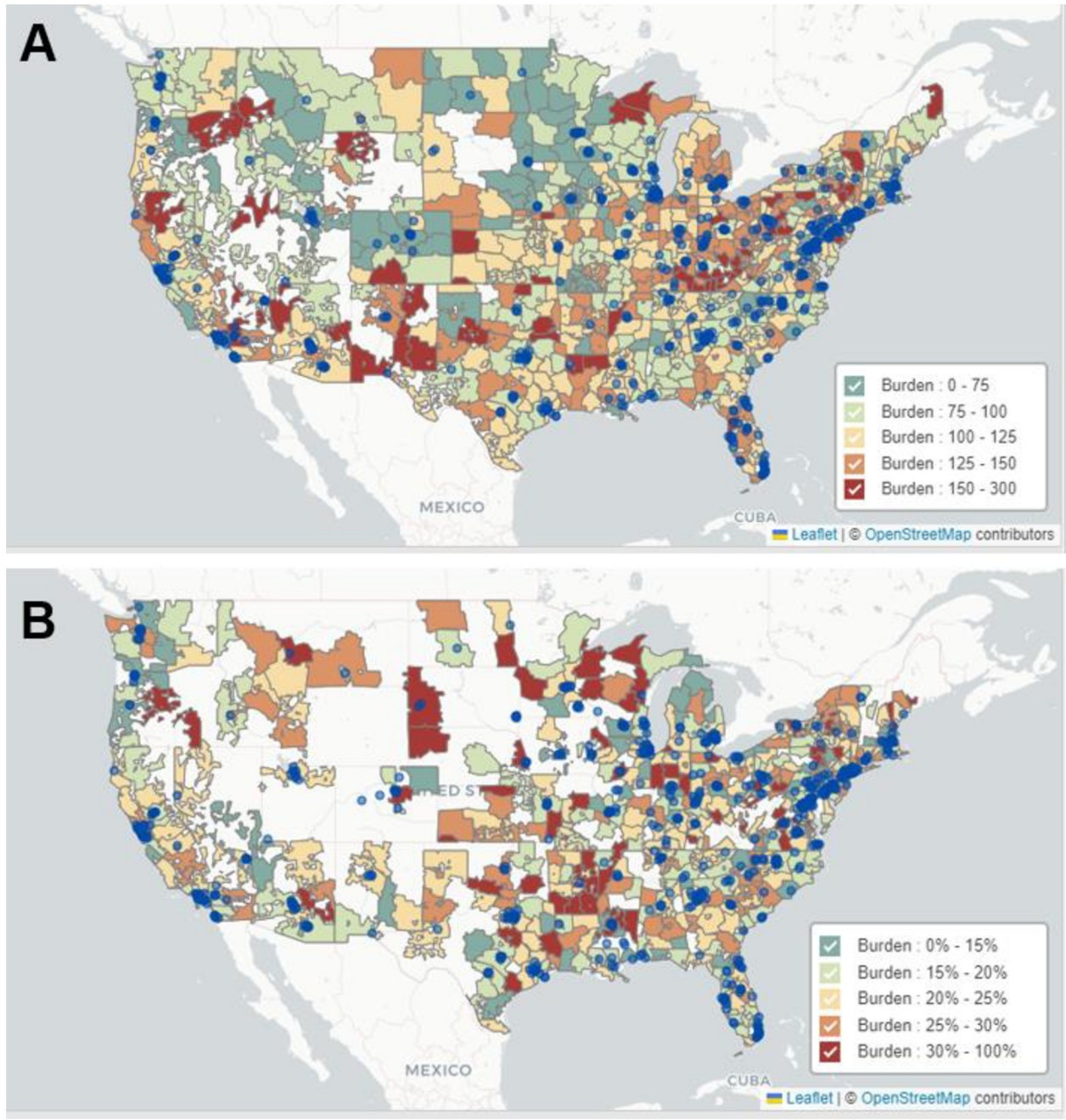

**Fig 1. Geographic variation in cervical cancer burden (A) and recurrent or metastatic cervical cancer burden (B) across the US for the period 2017–2022, and locations of brachytherapy centers as blue circles.** Cervical cancer burden was defined as the prevalent number of cervical cancer diagnoses per 100,000 eligible female enrollees. Recurrent or metastatic cervical cancer burden was defined as the proportion of patients with cervical cancer who initiated system therapy. Contains information from OpenStreetMap and OpenStreetMap Foundation, which is licensed under the Open Data Commons Open Database License (https://www.openstreetmap.org/copyright).

Effective screening for high-risk HPV and cervical precancerous lesions reduces incidence of CC and, among patients who are diagnosed at pre-cancerous or early stage, effective treatments with curative intent can reduce risk of recurrence [30–32]. The initial introduction of HPV vaccine in 2006 and expansion of the indicated population over the decades have

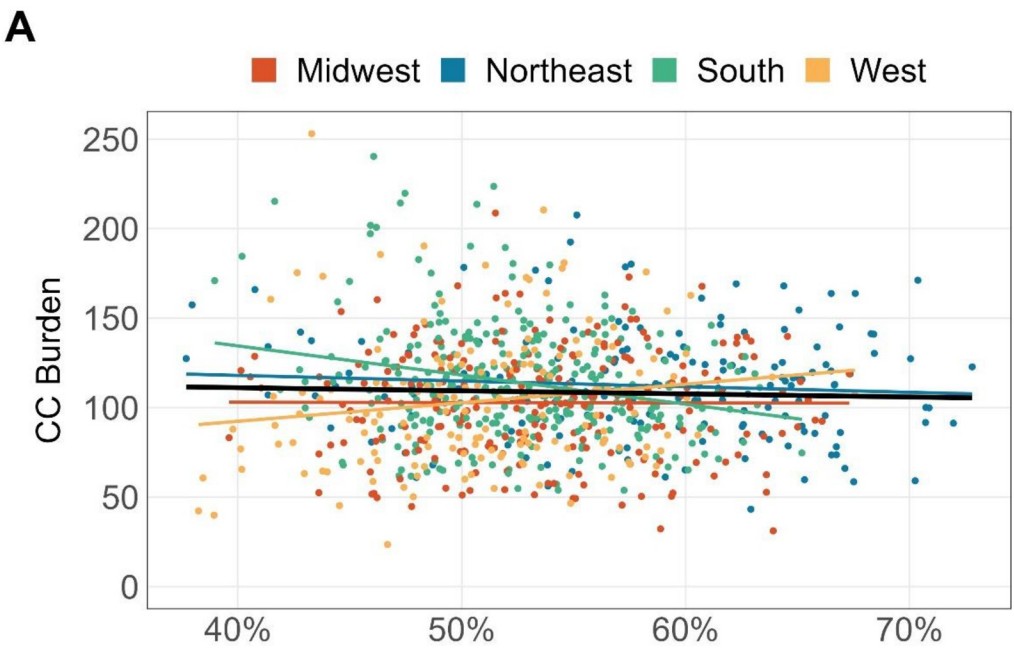

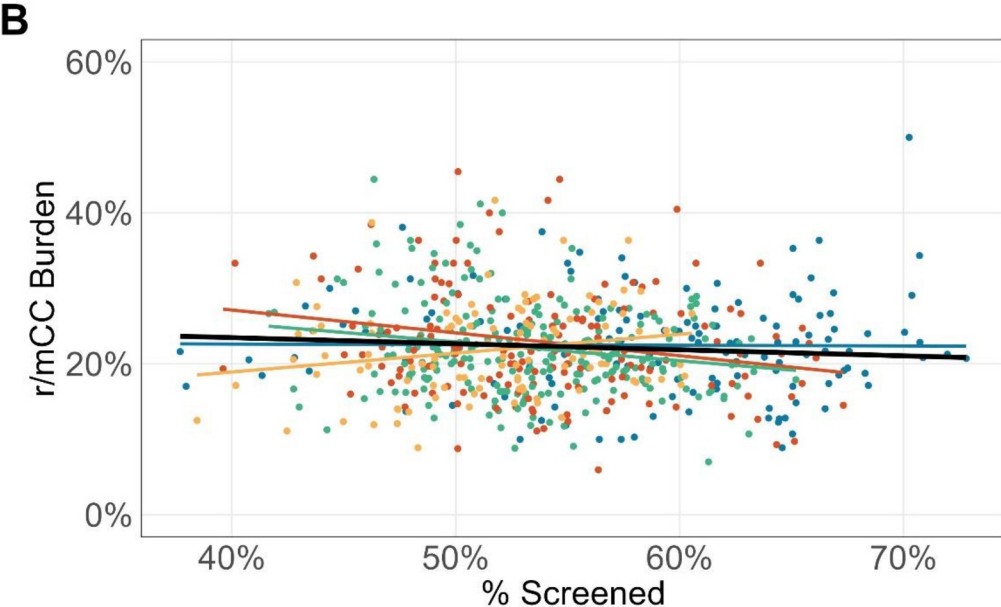

**Fig 2. Association between cervical cancer screening rate and cervical cancer burden (A) and recurrent or metastatic cervical cancer burden (B) at the ZIP-3 level for the period 2017–2022.** Cervical cancer burden was defined as the prevalent number of cervical cancer diagnoses per 100,000 eligible female enrollees. Recurrent or metastatic cervical cancer burden was defined as the proportion of patients with cervical cancer who initiated system therapy. CC: cervical cancer; r/mCC: recurrent or metastatic cervical cancer.

contributed to decrease in prevalence of the high-risk HPV infection which are more likely to lead to cervical cancer [33]. It is interesting to see that since the introduction of the vaccines, incidence and mortality rates of cervical cancer have remained relatively stable nationally [2]. This may suggest that there is a delayed effect in decreased risk of malignant disease due to the

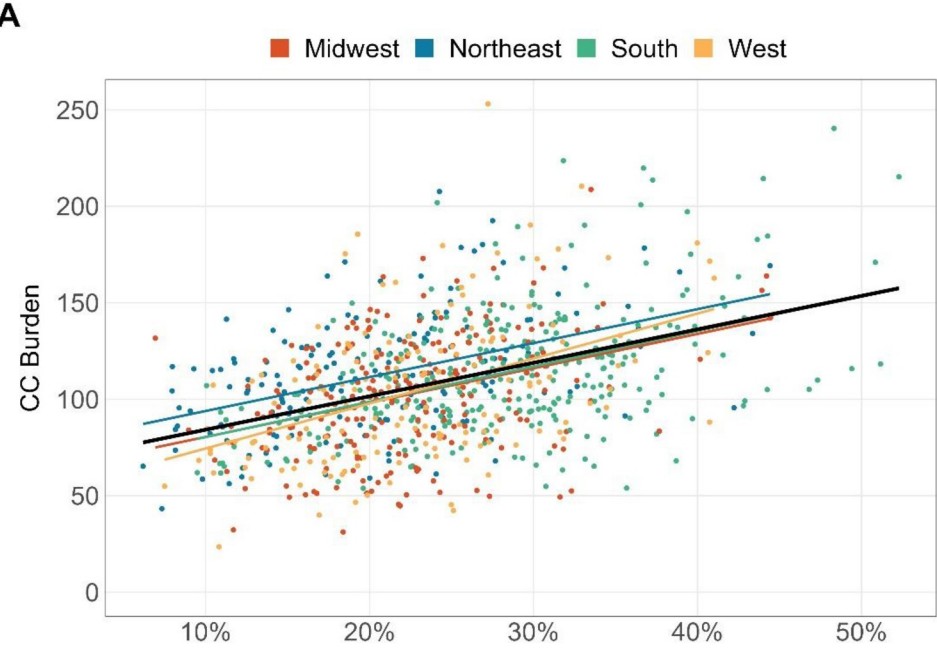

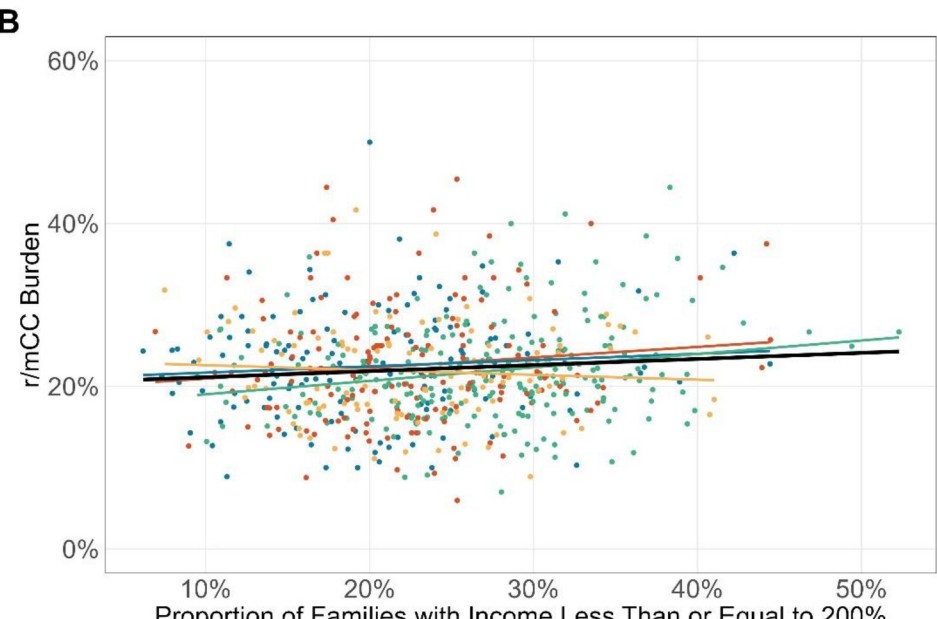

**Fig 3. Association between percentage of families under the 200% federal poverty level in a ZIP-3 and cervical cancer burden (A) and recurrent or metastatic cervical cancer burden (B) for the period 2017–2022.** Cervical cancer burden was defined as the prevalent number of cervical cancer diagnoses per 100,000 eligible female enrollees. Recurrent or metastatic cervical cancer burden was defined as the proportion of patients with cervical cancer who initiated system therapy. CC: cervical cancer; r/mCC: recurrent or metastatic cervical cancer.

| | CC Burden | | | | | r/mCC Burden | | | | |
|---|---|---|---|---|---|---|---|---|---|---|
| | All | Midwest | Northeast | South | West | All | Midwest | Northeast | South | West |
| % White | | ↘ | | ↗ | ↘ | | | | | |
| % Asian | ↘ | | | ↘ | | | ↘ | | | |
| % Black | | ↗ | ↗ | ↘ | | | | | | |
| % Hispanic | ↗ | | | | ↗ | | | | | |
| % Other | | ↘ | | | | | | | | |

**Fig 4. Association between the percentage of population belonging to a given race/ethnicity group in a ZIP-3 and cervical cancer burden and recurrent or metastatic cervical cancer burden for the period 2017–2022.** An arrow indicates a significant association between the race/ethnicity proportion and burden, with green upwards arrows indicating positive association and red downwards arrows indicating negative association. Cervical cancer burden was defined as the prevalent number of cervical cancer diagnoses per 100,000 eligible female enrollees. Recurrent or metastatic cervical cancer burden was defined as the proportion of patients with cervical cancer who initiated system therapy. CC: cervical cancer; r/mCC: recurrent or metastatic cervical cancer.

women at risk for cervical cancer today may not have been eligible for HPV vaccine at initial introduction, and that there are other risk factors, including sociodemographic status, which may confound interpretation of impact of HPV vaccination with malignant cervical cancer. Thus, in this study, we have focused on understanding association between the screening, rather than vaccination, and cervical cancer burden. While our univariate analysis showed higher screening rates generally corresponded with lower CC and r/mCC burden, West region had the opposite pattern of higher disease burden, suggesting insufficient effective preventative screening or inadequate treatment for early-stage disease.

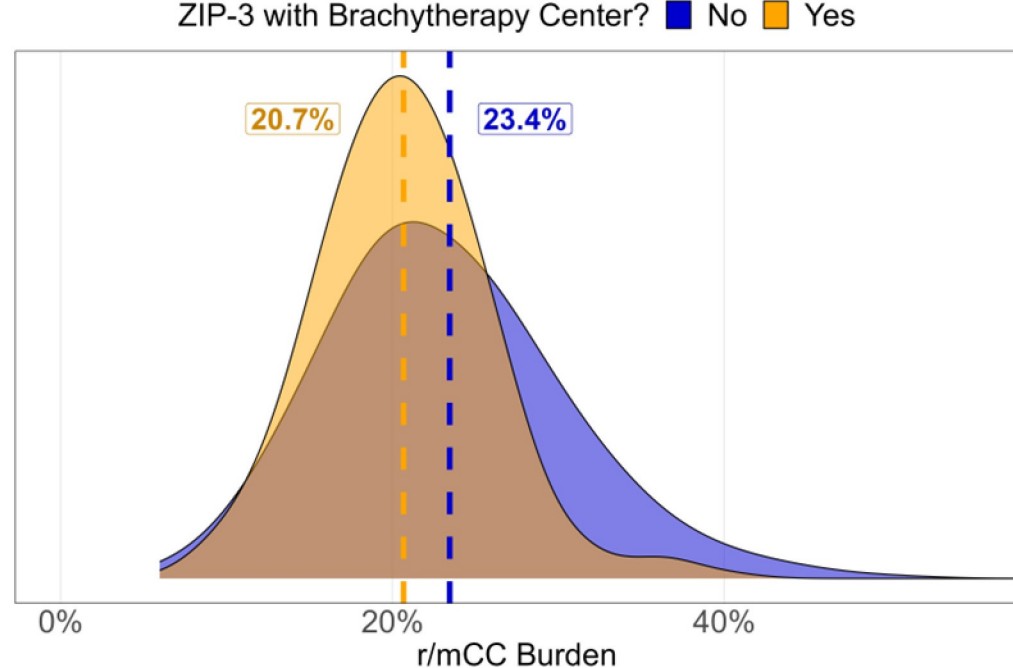

**Fig 5. Association between recurrent or metastatic cervical cancer burden and presence of at least one brachytherapy center in a ZIP-3 for the period 2017–2022.** Recurrent or metastatic cervical cancer burden was defined as the proportion of patients with cervical cancer who initiated system therapy. r/mCC: recurrent or metastatic cervical cancer.

Although there may be a myriad of potential factors contributing to the geographical variation in CC and r/mCC disease burden, the present study explored the impact on disease burden of previous cervical cancer screening, poverty, race/ethnicity, and availability of brachytherapy centers. These factors have been known to drive disparities in CC burden and outcomes [11–13]. We examined whether they also play a role in the observed geographical distribution of r/mCC burden and formed hypothesis as to other potential contributors of higher r/mCC burden.

Univariate associations between CC burden with sociodemographic risk factors in this study were generally consistent with expectation and that reported in literature [28, 34–37]. However, our finding of no significant association between proportion of low-income households and r/mCC burden at national level suggests that treatment for early-stage disease with curative intent is generally not influenced by poverty. The only region where a significant association was observed was in the South. We hypothesize that this may be explained by poverty in the South limiting patients' ability to access adequate treatment for early-stage disease, which was not as pronounced in other regions. Healthcare policies that enable patients diagnosed with early-stage CC to receive guideline-conforming care, such as The Breast and Cervical Cancer Treatment Program [38] available in many states, may help to alleviate healthcare disparity seen in the South in this study.

It is interesting to note that proportion of both Asian and Hispanic populations were significantly associated with overall CC burden, but that no specific race/ethnicity group showed significant association with r/mCC burden at national level. The authors interpret this observation as race/ethnicity do not affect either access or outcomes of early-stage disease treatment. Regional variations in CC burden observed in Fig 4 may suggest that education to raise awareness on preventative screening and early medical intervention after abnormal screening result should be implemented consistently in our efforts to eradicate CC. The authors note that while previous literature suggest Black and Hispanic female populations experience higher incidence and mortality rates of cervical cancer across the US, this finding is generally consistent with the literature on the suggested sociodemographic factors that may contribute to the observed race/ethnic disparity in outcomes [8].

Lastly, in this study, we mapped locations of brachytherapy centers, as a proxy for availability of standard-of-care (SOC) treatment with curative intent. The visual map (Fig 1) showed that some patients may have to travel substantial distances to a SOC treatment center, especially in the West. Given that a meaningful proportion of CC patients are of low socioeconomic status, travel requirements may become prohibitive for them to access life-saving treatments. While it is resource-intensive to establish specialty centers near all patients, we believe that provisions to support patients with transportation or basic logistical needs may offer a short-term solution. Lastly, further contextualization of these insight and interpretation may be enabled by a future comparison of the CC patient population across US geographies using established indices of healthcare access, such as the Healthcare Access and Quality index [39].

The present study has several limitations. First, as with any analysis utilizing administrative claims data, caution should be exercised when attempting to generalize study findings beyond the insured population. Second, since the current study aimed to generate hypotheses about potential risk factors of observed regional variation in CC and r/mCC burden to guide future research directions, only univariate analyses were performed to facilitate initial interpretation at the most granular geographical level possible. As such, results may be confounded and should be interpreted with caution. Third, as the analyses were performed at the ZIP-3-level instead of the patient-level, one must be careful when extrapolating these findings to individuals. Fourth, the need to censor data with ≤3 patients may result in potential underrepresentation of r/mCC burden in some geographies. Finally, our analyses were limited to variables

with ZIP-3-level data, and thus may have excluded potential relevant contributors of the geographic disparity in disease distribution.

## Conclusion

CC and r/mCC disparities are linked to certain socioeconomic factors, including poverty level, race/ethnicity, and access to modern early-stage treatment. The study also visualized the variability of disease burden among US geographical regions to highlight areas of need for healthcare providers and policy makers, as well as the broader healthcare community. Hypotheses generated from this study support recommendations to pinpoint gaps in the cervical cancer care continuum that may be contributing to disparities in disease burden, such as in the West, where higher screening rates did not translate to lower r/mCC burden. Overall, findings from this study may help to optimize healthcare resources allocation, and advocacy and education of modern treatment options to minimize disparities in outcomes for patients in the US.

## Supporting information

**S1 Table. States in each region.**
(DOCX)

**S2 Table. Annual prevalent cervical cancer burden (cases per 100,000 women), recurrent or metastatic cervical cancer burden (proportion of patients with cervical cancer who initiated systemic therapy) and proportion of eligible population screened from 2017–2022.**
(DOCX)

**S3 Table. Fitted univariate regression coefficients, confidence intervals, standard errors, p-values, and $R^2$ values for univariate regressions stated in the main text.**
(DOCX)

**S1 Fig. Geographic variation in cervical cancer screening rate across the US for the period 2017–2022, and locations of brachytherapy centers as blue circles.** A patient was considered screened for cervical cancer during a year of interest if she was either between the ages of 21 and 64 and had cervical cytology performed within the previous three years, or between the ages of 30 and 64 and had cervical hrHPV testing performed with or without cytology within the previous five years. Contains information from OpenStreetMap and OpenStreetMap Foundation, which is licensed under the Open Data Commons Open Database License (https://www.openstreetmap.org/copyright).
(TIF)

## Acknowledgments

The authors thank Xinshuo Ma and Kristina Cheng from Komodo Health for their help in identifying CC and r/mCC patients from the Komodo Healthcare Map database.

## Author Contributions

**Conceptualization:** Tara Castellano, Christina Washington, Jie Ting, Yitong J. Zhang, Fernanda Musa, Kathleen Moore, Leslie Randall, Jagpreet Chhatwal, Turgay Ayer, Charles A. Leath, III.

**Data curation:** Andrew K. ElHabr, Ezgi Berksoy.

**Formal analysis:** Andrew K. ElHabr.

**Investigation:** Tara Castellano, Andrew K. ElHabr, Jie Ting, Yitong J. Zhang, Ezgi Berksoy, Turgay Ayer, Charles A. Leath, III.

**Methodology:** Tara Castellano, Jie Ting, Yitong J. Zhang, Turgay Ayer, Charles A. Leath, III.

**Supervision:** Tara Castellano, Jie Ting, Yitong J. Zhang, Turgay Ayer, Charles A. Leath, III.

**Validation:** Tara Castellano, Andrew K. ElHabr, Jie Ting, Yitong J. Zhang, Ezgi Berksoy.

**Visualization:** Andrew K. ElHabr.

**Writing – original draft:** Tara Castellano, Andrew K. ElHabr, Jie Ting, Yitong J. Zhang.

**Writing – review & editing:** Christina Washington, Fernanda Musa, Kathleen Moore, Leslie Randall, Jagpreet Chhatwal, Turgay Ayer, Charles A. Leath, III.

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
