## [Decision Letter · Decision Letter 0]

17 May 2024

PONE-D-24-16450Health Disparities in Cervical Cancer: Estimating Geographic Variations of Disease Burden and Association with Key Socioeconomic and Demographic Factors in the USPLOS ONE

Dear Dr. ElHabr,

Thank you for submitting your manuscript to PLOS ONE. After careful consideration, we feel that it has merit but does not fully meet PLOS ONE’s publication criteria as it currently stands. Therefore, we invite you to submit a revised version of the manuscript that addresses the points raised during the review process.

We look forward to receiving your revised manuscript.

Kind regards,

Sina Azadnajafabad, MD, MPH

Academic Editor

PLOS ONE

Journal Requirements:

"Tara Castellano has received consulting fees from Glaxo-Smith-Klein; Andrew K. ElHabr and Ezgi Berksoy are paid employees of Value Analytics Labs, a healthcare consultancy company; Jie Ting and Yitong J. Zhang are employees and stack holders of Pfizer Inc.; Kathleen Moore has participated in data monitoring or advisory boards for Astra Zeneca, Aravive, Alkemeres, Aadi, Blueprint pharma, Clovis, Caris, Duality, Elevar, Eisai, EMD Serono, GSK/Tesaro, Genentech/Roche, Hengrui, Immunogen, Janssen, Lilly, Mersana, Merck, Myriad, Mereo, Novartis, OncXerna, Onconova, SQZ, Tarveda, VBL Therapeutics, Verastem and Zentalis, received support for attending meetings from Astra Zeneca, GSK/Tesaro, and BioNTech, and holds a leadership role with GOG partners; Leslie Randall reports personal fees from Seagen Inc. for an educational webinar at drug launch and speaker's bureau and personal fees from Merck for an unbranded educational video for cervical cancer. Her institute receives research funding for clinical research from Seagen and Merck. She reports personal fees from BluPrint Oncology, PER, CurioScience, Projects in Knowledge, AstraZeneca, Tesaro, Merck, Mersana, Agenus, Rubius Therapeutics, Myriad Genetics, EMD Serono, Genentech/Roche, Seattle Genetics, Novartis, and Eisai, all outside the submitted work; Jagpreet Chhatwal and Turgay Ayer are co-owners of Value Analytics Labs; Charles A. Leath III has received consulting fees from Seagen Inc. for service on Scientific Advisory boards, cervical cancer research funding from Agenus, Rubius Therapeutics, and Seagen Inc., and funding from the NCI UG1 CA23330 and P50 CA098252; Fernanda Musa and Christina Washington have no competing interests to disclose."

4. We note that Figure 1 and S1 in your submission contain map/satellite images which may be copyrighted. All PLOS content is published under the Creative Commons Attribution License (CC BY 4.0), which means that the manuscript, images, and Supporting Information files will be freely available online, and any third party is permitted to access, download, copy, distribute, and use these materials in any way, even commercially, with proper attribution. For these reasons, we cannot publish previously copyrighted maps or satellite images created using proprietary data, such as Google software (Google Maps, Street View, and Earth). For more information, see our copyright guidelines: http://journals.plos.org/plosone/s/licenses-and-copyright.

a. You may seek permission from the original copyright holder of Figure 1 and S1 to publish the content specifically under the CC BY 4.0 license.  

Reviewers' comments:

Reviewer's Responses to Questions

**Comments to the Author**

1. Is the manuscript technically sound, and do the data support the conclusions?

Reviewer #1: Yes

Reviewer #2: Partly

2. Has the statistical analysis been performed appropriately and rigorously? 

Reviewer #1: Yes

Reviewer #2: Yes

3. Have the authors made all data underlying the findings in their manuscript fully available?

Reviewer #1: Yes

Reviewer #2: Yes

4. Is the manuscript presented in an intelligible fashion and written in standard English?

Reviewer #1: No

Reviewer #2: Yes

5. Review Comments to the Author

Reviewer #1: • In the abstract and introduction, it would be more informative if the authors could elaborate on what they mean by “burden”? in the eyes of epidemiologists, the burden could be incidence, prevalence, mortality, DALY, or a combination of them.

• The storyline is not good through the introduction and results. The manuscript is not written in a readable style. The study has valuable data. However, this data has not been correctly directed to the objective of the study, which is to be a guide for policymakers.

• Please add some previous studies reporting the possible associated factors with the disparities in CC so the reader can understand the base of your hypotheses. Or, you can provide references for line 105.

• Results, line 176: It is not clear why “14,033 initiated a systemic therapy treatment” equals r/mCC burden”

• Results, the authors have the data for both screenings and sociodemographics. I wonder why they haven’t analyzed their association which would be a valuable output for this study.

• Discussion: I was anticipating that the authors would analyze the association between HPV vaccination and CC burden across the country, but it was missing. This result would demonstrate the lack or sufficiency of this preventive measure in the shadow of the sociodemographic status.

• Discussion: there are some previously established indices for assessing the quality of care such as healthcare access and quality (HAQ) (28528753) or quality of care index (QCI) (38273304). It is necessary for the authors to compare and justify their methodology and results (if applicable) with these indices.

Reviewer #2: Thank you for the opportunity to review this study. The authors leveraged claims and US Census Bureau data to assess the geographic variations in cervical cancer and poor outcomes (recurrence and metastasis) burden. They also examined the associations between cervical cancer screening rates and key variables in relation to cervical cancer burden. The study was well-designed and has interesting findings. However, I do have some questions/suggestions for improvement below.

Major issues

Introduction: The introduction is superficial and lacks information on the rationale and justification for the study. Could the author elaborate more on what previous studies have done, current gaps, and what the current study aims to add to knowledge on this specific issue?

Introduction: Please define brachytherapy at first use, for the benefit of readers who may not be familiar with the term. Also, include citations of studies that have used brachytherapy as proxy for access to guideline-concordant therapies

Methods: Could the authors expand on the Komodo Healthcare Map and what patient data are entered/included and any standardized processes guiding data inclusion? Data stated in line 109 (and presented in Table 1) is 2015-2022 different than the analyzed data 2017-2022. I recommend presenting data specific to this analysis.

Methods: It appears some information is missing in lines 123 “with a diagnosis for malignant neoplasm of the cervix as identified by the ICD-123 9-CM code 180.xx or ICD-10-CM code C53.xx.”

Methods: Please elaborate on these units of analysis and why you transformed/aggregated ZIP-5 to ZIP-3 “and race/ethnicity at the ZIP-5 level, which were then aggregated to the ZIP-3 level (6)”. Also, what proportion of enrollees were excluded due to missing ZIP-3 information?

Results: The results section needs a major revision (Lines 197-261). Many sentences are fragmented. For example, figure labels are used in presenting results and copied verbatim in tables/ figures at the end of the manuscript.

Results: Table 1 shows data from 2015-2022 and study analyzed data from 2017 to 2022. It would be less confusing if the authors presented data specific to their analysis.

Discussion: The authors should provide additional details by comparing their findings with past studies and substantiating their claims using extant data.

Discussion: The authors introduced a new term ‘social determinants’ in the conclusion without a single mention in the preceding sections. I recommend sticking with the terms you have used throughout ‘socioeconomic’ and avoid introducing a new one in the closing paragraph.

Minor issues

Abstract: use ‘approximately’ instead of > unless there is a word count issue here.

Results: For the maps, can you swap the green shades such that Burden 0-15% is dark green and 15-20% is light green (similar to the dark red that signifies more serious problem)? This would enhance the visibility of the data on the maps.

Results: Figures and tables are stand-alone data without a need to refer to the body of the manuscript to understand the data presented. Hence, would you please provide footnotes for each figure (and tables) offering additional important information to understand the data being presented?

Results and Discussion: The manuscript’s readability can be improved by organizing the results and discussion logically.

6. PLOS authors have the option to publish the peer review history of their article (what does this mean?). If published, this will include your full peer review and any attached files.

Reviewer #1: **Yes: **Mohammadreza Azangou-Khyavy

Reviewer #2: No

---

## [Author Response · Author response to Decision Letter 0]

12 Jun 2024

Please see the Responses to Reviewers file for all of our responses. Thank you again to the reviewers for their comments, which we believe have increased the quality of this manuscript.

---

## [Decision Letter · Decision Letter 1]

3 Jul 2024

Health Disparities in Cervical Cancer: Estimating Geographic Variations of Disease Burden and Association with Key Socioeconomic and Demographic Factors in the US

PONE-D-24-16450R1

Dear Dr. ElHabr,

We’re pleased to inform you that your manuscript has been judged scientifically suitable for publication and will be formally accepted for publication once it meets all outstanding technical requirements.

Kind regards,

Sina Azadnajafabad, MD, MPH

Academic Editor

PLOS ONE

Additional Editor Comments (optional):

Reviewers' comments:

Reviewer's Responses to Questions

**Comments to the Author**

1. If the authors have adequately addressed your comments raised in a previous round of review and you feel that this manuscript is now acceptable for publication, you may indicate that here to bypass the “Comments to the Author” section, enter your conflict of interest statement in the “Confidential to Editor” section, and submit your "Accept" recommendation.

Reviewer #1: All comments have been addressed

Reviewer #2: All comments have been addressed

2. Is the manuscript technically sound, and do the data support the conclusions?

Reviewer #1: Yes

Reviewer #2: Yes

3. Has the statistical analysis been performed appropriately and rigorously? 

Reviewer #1: I Don't Know

Reviewer #2: Yes

4. Have the authors made all data underlying the findings in their manuscript fully available?

Reviewer #1: Yes

Reviewer #2: Yes

5. Is the manuscript presented in an intelligible fashion and written in standard English?

Reviewer #1: Yes

Reviewer #2: Yes

6. Review Comments to the Author

Reviewer #1: Thanks for addressing the comments. The study intended to investigate the disparities in cervical cancer prevalence in the US. Now, the manuscript seems sound, and the discussion and introduction support the results part. The method section has also been revised and is more comprehensive.

Reviewer #2: The authors have addressed many of the concerns raised and the manuscript's readability is much improved. Thank you.

7. PLOS authors have the option to publish the peer review history of their article (what does this mean?). If published, this will include your full peer review and any attached files.

Reviewer #1: No

Reviewer #2: No

---

## [Editor Report · Acceptance letter]

9 Jul 2024

PONE-D-24-16450R1 

PLOS ONE

Dear Dr. ElHabr, 

I'm pleased to inform you that your manuscript has been deemed suitable for publication in PLOS ONE. Congratulations! Your manuscript is now being handed over to our production team.

Kind regards, 

on behalf of

Dr. Sina Azadnajafabad 

Academic Editor

PLOS ONE